# OBJECT-CENTRIC COMPOSITIONAL IMAGINATION FOR VISUAL ABSTRACT REASONING

**Rim Assouel**[1,2]**, Pau Rodriguez**[2]**, Perouz Taslakian**[2,3]**, David Vazquez**[2]**, Yoshua Bengio**[1]
[1]Mila, Universite de Montreal; [2]Servicenow Research [3]Samsung AI Montreal
Correspondance : `assouelr@mila.quebec`

*Imagination will often carry us to worlds that never were.*
*But without it we go nowhere — Carl Sagan.*

## ABSTRACT

Like humans devoid of imagination, current machine learning systems lack the ability to adapt to new, unexpected situations by foreseeing them, which makes them unable to solve new tasks by analogical reasoning. In this work, we introduce a new compositional imagination framework that improves a model's ability to generalize out-of-distribution. One of the key components of our framework is object-centric inductive biases that enables models to perceive the environment as a series of objects, properties, and transformations. By composing these key ingredients, it is possible to generate new unseen tasks that, when used to train the model, improve systematic generalization. Experiments on a simplified version of the *Abstraction and Reasoning Corpus (ARC)* demonstrate the effectiveness of our framework.

## 1 INTRODUCTION

Using simple concepts as building blocks, humans have the capacity to compose and enhance knowledge by relating it to previous experiences. Such analogical reasoning [23, 21] leverages the compositional structure of the world to make sense of new experiences and to imagine previously unseen scenarios. Many machine learning models are also able to acquire knowledge from data and use it successfully to perform a given task. However, their ability to adapt this knowledge to new domains remains unsatisfactory [4].

Objects are one of the core abstractions of the human brain when applying analogical reasoning [23]. For instance, we can infer the properties of a new object by transferring our knowledge of these properties from similar objects [21]. This realization has inspired a recent body of work that focuses on learning models that discover objects in a visual scene without supervision [8, 17, 10, 27, 11, 3, 28, 20]. Many of these works propose several inductive biases that lead to a visual scene decomposition in terms of its constituting objects. The expectation is that such object-centric decomposition would lead to better generalization since it better represents the underlying structure of the physical world [22]. To the best of our knowledge, the effect of such object-centric representations for systematic generalization in visual reasoning tasks remains largely unexplored.

In an attempt to better measure the gap between machine and human learning, Chollet [4] introduced the *Abstraction and Reasoning Corpus (ARC)*. ARC consists of visual analogy tasks made of few input-output pairs. The goal of a model is to learn to infer the program applied to the inputs from few demonstrations and test it on novel samples. The ARC challenge remains insurmountable by current ML systems. Chollet [4] hypothesizes that a successful algorithm should be able to synthetize new solutions by recombining a set of subprograms or primitives based on core knowledge [23].

Dreamcoder [7] is one of the methods that are closest to Chollet's desiderata for solving ARC. In their work, Ellis et al. [7] propose imagining new program instances by composing primitives from a domain-specific language (DSL) and training the model to recognize the resulting solutions. In their work, the authors show that augmenting the training data with imagined programs is critical in

a low data regime. They also test their model on a suite of program induction domains where the developer can anticipate which primitives it needs to provide in the initial DSL. In a visual domain such as ARC, however, such primitives require more high-level notions related to core knowledge and are difficult to anticipate.

In this work, we propose a way of using object-centric inductive biases to derive a new *compositional imagination* framework in which a model can learn a series of concepts without the need for specifying a DSL. We show that by imagining new scenarios that are composed of learned concepts, and by learning to predict back those concepts, the model better generalizes to compositions unseen during training. Finally, since exploring new methods directly on ARC is a daunting task, we introduce Sort-of-ARC, a dataset where the programs operate on objects and are conditioned on some of their attributes in a controlled environment.

## 2  RELATED WORK

**Visual Reasoning.**  Both Hoshen & Werman [13] and Barrett et al. [2] propose to assess the reasoning ability of a neural network by applying standardized intelligence tests such as Raven's Progressive Matrices (RPM) [15, 35]. These kinds of problems have inspired a large number of deep learning methods [12, 14, 24, 38, 31, 34, 36]. In a recent survey, Mitchell [21] recommends evaluating models on generative tasks that focus on human core knowledge [23]. ARC presents one such task, where rather than choosing an answer, the solver has to generate its own solution, and hence is more resistant to learning shortcuts (unlike in the case of RPM). However, ARC remains unapproachable by current deep learning methods. Since its release, the *neural abstract reasoner* [16] is the only deep learning method that succeeds in a reduced subset of ARC's problems.

**Object-centric Representations.**  The objective of this line of work is to represent a visual scene in terms of the objects that compose it. Spatial mixture models [20, 3, 11] define a Gaussian mixture model weighted by the slot masks decoded from a set of learned structured latents. Spatial-transformer-based models further disentangle each slot's latent representation into several variables (e.g. content, location, presence, depth) [8, 17, 5, 25, 19]. All these approaches focus on the generative abilities of the models; in our case, we study the impact of object-centric inductive biases on systematic generalization of the models in a visual reasoning task. We observe that modularity of representations is as important as the mechanisms [9] that operate on them. Additionally, we show that object-centric inductive biases of both representations and mechanisms allow us to derive an imagination framework that leads to better systematic generalization.

**Imagination in Deep Learning.**  It is well known that data generation techniques such as data augmentation [1, 6, 32], domain randomization [26], and mixup [37] can be used to improve the generalization capacity of a model. These methods tend to generate samples that lie in the training set manifold. Thus, they are not adequate for algorithmic reasoning tasks that that evaluate models on completely novel situations. Another body of work leverages the compositional structure of the world and of algorithms to generate samples that could not be obtained by interpolation [18, 33, 7]. However, the latter works rely on a set of defined primitives or DSL, which is not possible to obtain in more realistic or challenging settings.

## 3  OBJECT-CENTRIC COMPOSITIONAL IMAGINATION

We propose a model that leverages compositional inductive biases in the architecture to generate new data points. Our model learns to compose object-centric representations with a series of learned operations, and generate new imagined scenarios by re-arranging these components in novel ways. Training models on imagined scenarios helps these models to generalize more systematically.

### 3.1  SORT-OF-ARC

While the performance of a model on ARC is an interesting metric for comparing artificial reasoning systems, ARC's level of difficulty makes it impenetrable from a research perspective. We thus introduce Sort-of-ARC, a toy version of ARC that shares the same input space and presents a series

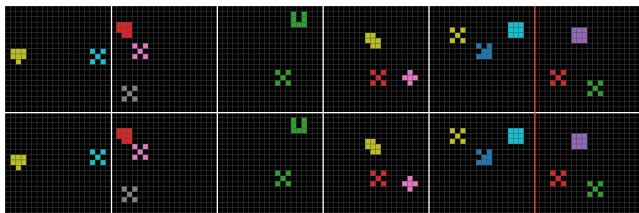

Figure 1: **Sort-of-ARC episode.** The top row contains the inputs and the bottom row the outputs after applying a program on the input. In the figure, the program moves shape X one pixel up. The model receives the first five columns as support set and the input (top) of the sixth column as query. The output of the model is compared to the output of the query (bottom).

of simpler problems. These problems are generated by systematically applying a series of operations that test different aspects of reasoning algorithms, and which might eventually be extended and scale to the full benchmark. Some of these aspects include the ability to make abstractions such as objects, retrieving these objects based on their properties, applying operations to objects, and generalizing to new unseen combinations of objects, properties, and operations.

For each task, the model is given a support set indexed by $\mathbf{S}$ composed of visual I/O pairs $\{\mathbf{x}_i, \mathbf{y}_i\}_{i \in \mathbf{S}}$ with $\mathbf{x}_i, \mathbf{y}_i \in \mathbf{R}^{d \times d \times c}$. A task transforms each input of the same support set according to a program that is defined by a *condition* and a *transformation*. The condition selects the object properties to which we will apply a particular transformation (e.g., the condition/transformation pair red/right translation would consist in translating all the red objects to the right).

During training, the model is presented with a support set of N I/O pairs $\{\mathbf{x}_i, \mathbf{y}_i\}$ to which we apply the same condition/transformation rule and train it to reconstruct the corresponding output $\mathbf{y}_q$ of query input $\mathbf{x}_q$. The model has to first *infer the underlying program* that generated the support set and then *apply* it to a query input to generate the right output. To explicitly measure the systematic generalization abilities of object-centric models, we consider an out-of-distribution (OOD) train/test split setting where all the *condition* and *transformation* primitives are seen during training individually, but some pairs are held-out in the test set. More details can be found in the Appendix A.

## 3.2 OUR MODEL

Our model is composed of two parts, a **controller** and an **executor**. The controller encodes the support set and outputs an embedding containing *instructions*; the executor then modifies an input query conditioned on these instructions. All our model variants and baselines differ from one another by the executor architecture. We provide a schematic of our model in Figure 5 and more details in Algorithm 1.

### 3.2.1 CONTROLLER

The controller encodes a set of pairs $\{\mathbf{x}_i, \mathbf{o}_i\}_{i=1..N}$ of inputs X and outputs O, and outputs an instruction embedding $\mathbf{z}$ that will be given to the executor. We consider different architectures for the controller: The perception backbone can either be slot-attention or CNN-based that would encode each image in the support set. The backbone is then followed by a transformer encoder [29] to output the instruction embedding $\mathbf{z} \in \mathbb{R}^p$ given to the executor.

### 3.2.2 EXECUTOR

The executor takes as input a query $\mathbf{x}_q \in \mathbb{R}^{d \times d \times c}$ and an instruction embedding $\mathbf{z} \in \mathbb{R}^p$ provided by the controller, and outputs the resulting query modification $\mathbf{o}_q \in \mathbb{R}^{d \times d \times c}$. This process is composed of four main steps: (1) Decomposing the visual query input $\mathbf{x}_q$ into a set of $K$ entity-centric latent representations $\mathbf{H} = \{\mathbf{h}_k\}_{k=1..K}$. (2) Translating the instruction embedding $\mathbf{z}$ into a *neural program* via a *selection bottleneck* (details below). (3) Structured update of the the entity-centric latent query set $\mathbf{H}$ according to the selected *neural program*. (4) Entity-centric rendering of the updated latent set. Steps (1) and (4) are parametrized by a slot attention module [20] and a spatial broadcast

decoder [30], respectively. The novelty of our approach lies in Steps (2) and (3), which we describe in detail below.

**Neural Program Selection Bottleneck.** The executor translates an instruction embedding $\mathbf{z}$ into a neural program via a key-query attention mechanism. This mechanism selects a (1) *condition* $\mathbf{c}$, which is used to select which latent slots the program should update; and a (2) *transformation* $\mathbf{p}$, which defines how the latent slots should be modified. We let $(\mathbf{c}, \mathbf{c}_{\text{prob}}), (\mathbf{p}, \mathbf{p}_{\text{prob}}) = \text{Selection}(\mathbf{z})$ be the output of this selection mechanism, with $\mathbf{c}_{prob}, \mathbf{p}_{prob}$ the selection probabilities obtained by the soft attention mechanism described below.

The queries of the two soft attention mechanisms are extracted from the instruction embedding $\mathbf{z}$ with two different parametric operations, $f_{\text{query}}^c$ and $f_{\text{query}}^p$, such that $Q_c = f_{\text{query}}^c(\mathbf{z}) \in \mathbb{R}^{1 \times p}$ and $Q_p = f_{\text{query}}^p(\mathbf{z}) \in \mathbb{R}^{1 \times p}$ where $c$ and $p$ are used to index the condition and the transformation selection mechanisms, respectively. The keys and values are partitioned and learned embeddings as part of the model and we denote them as $K_c, V_c \in \mathbb{R}^{N_c \times p}$ and $K_p, V_p \in \mathbb{R}^{N_p \times p}$, where $N_c$ (resp. $N_p$) denotes the number of learned condition (resp. transformation) embeddings. The resulting neural program is thus described by the softly selected condition $\mathbf{c}$ and transformation $\mathbf{p}$ embeddings obtained by the usual key-query attention mechanism:

$$\mathbf{c} = \text{softmax}(\frac{Q_c K_c^T}{\sqrt{p}}) V_c \quad \text{and} \quad \mathbf{p} = \text{softmax}(\frac{Q_p K_p^T}{\sqrt{p}}) V_p \tag{1}$$

**Entity-Centric Update.** Given a condition $\mathbf{c}$ and a transformation $\mathbf{p}$, the latent slots $\mathbf{H}$ extracted from a query image are updated in an entity-centric way. This update mechanism produces a latent output $\mathbf{H}^{new} = \text{Update}(\mathbf{H}, \mathbf{c}, \mathbf{p})$.

The condition $\mathbf{c}$ selects which slot to update by defining an update *gate* for each slot; for a given slot $k$, the update gate $\alpha_k$ is obtained by comparing it to the condition such that $\alpha_k = \sigma(\text{MLP}_{\text{pres}}([\mathbf{h}_k, \mathbf{c}]))$. The transformation $\mathbf{p}$ defines a slot-wise update $\tilde{\mathbf{h}}_k = \text{MLP}_{\text{up}}([\mathbf{h}_k, \mathbf{p}])$ such that updated slot representation for slot $k$ is defined by $\mathbf{h}_k^{\text{new}} = \mathbf{h}_k + \alpha_k * \tilde{\mathbf{h}}_k$. The output image is then rendered using a Spatial Broadcast Decoder [30] on the new slots $\mathbf{H}^{new}$.

We let $\mathbf{o} = \text{Decode}(\mathbf{H}^{new})$ be the output image obtained by the structured entity-centric update to the query image $\mathbf{x}$, conditioned on condition and transformation embeddings $\mathbf{c}$ and $\mathbf{p}$.

### 3.3 COMPOSITIONAL IMAGINATION

Our main contribution consists of showing how object-centric inductive biases enable a learning paradigm in which new tasks are imagined and composed of primitive concepts learned during training. Similar to the work by Ellis et al. [7], this paradigm is loosely inspired by the wake-sleep algorithm in which a recognition model (e.g., our controller) is trained on imagined data produced by a generative model (e.g., our executor). We propose to imagine new scenarios by selecting, in a random and uniform way, a condition $i_c$ and a transformation $i_p$. We then train the model to predict the proper condition and transformation indices given the imagined I/O set. During the imagination phase, we reuse the episodes seen during the "wake" phase $\mathbf{X} = \{\mathbf{x}_i\}_{i=1..N}$. The imagination phase is thus composed of the following steps:

$$i_c \sim U(1, N_c), i_p \sim U(1, N_p), \tag{2}$$

$$\mathbf{o}_i^{i_c, i_p} = \text{Decode}(\text{Update}(\mathbf{h}_i, \mathbf{c}_{i_c}, \mathbf{p}_{i_p})), \quad \mathbf{z}^{i_c, i_p} = \text{Controller}(\mathbf{X}, \mathbf{O}^{i_c, i_p}) \tag{3}$$

$$(\mathbf{c}^{i_c, i_p}, \mathbf{c}_{\text{prob}}^{i_c, i_p}), (\mathbf{p}^{i_c, i_p}, \mathbf{p}_{\text{prob}}^{i_c, i_p}) = \text{Selection}(\mathbf{z}^{i_c, i_p}). \tag{4}$$

The model is trained to predict which condition/transformation were sampled to construct the imagines episodes and thus minimize the following cross-entropy loss:

$$\mathcal{L}^{\text{im}} = -\log(\mathbf{c}_{\text{prob}}^{i_c, i_p})_{i_c} - \log(\mathbf{p}_{\text{prob}}^{i_c, i_p})_{i_p} \tag{5}$$

where $(\mathbf{c}_{\text{prob}}^{i_c, i_p})_{i_c}$ is the $i_c$ (and $i_p$ respectively) position of predicted probability vector for the inferred condition $\mathbf{c}^{i_c}$.

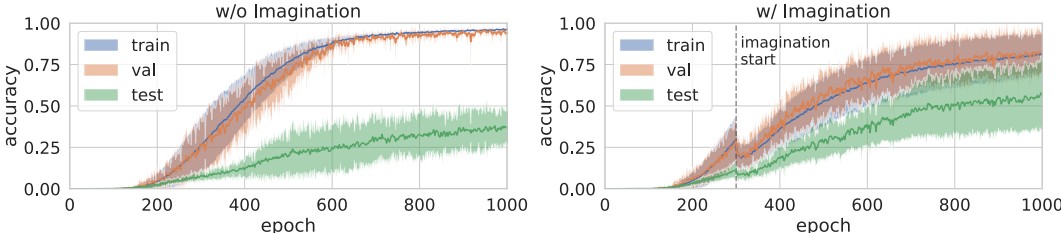

Figure 2: Train, validation, and test accuracy curves for our model when trained without (left) and with (right) imagined episodes. Imagined episodes are introduced after epoch 300.

The overall training objective is thus composed of the query output prediction loss and the imagination loss. Details about the coefficients and different warm-up schedules are given in the Appendix C.

## 4 EXPERIMENTS

Our main goal is to showcase that object-centric inductive biases allow us to derive a *compositional imagination* training framework that help with systematic generalization. Thus, we address the following question: Does training on imagined situations improve generalization on new unseen situations composed of known training concepts? To do so, we propose a number of baselines and model variations that specifically measure the importance of, on the one hand, object-centric inductive biases in the representations and/or the execution and, on the other hand, the compositional imagination framework that we propose.

**Baselines.** In order to verify whether object-centric representations are enough for systematic generalization to emerge we consider a baseline where we remove the selection bottleneck and the mechanism that operate on slots is thus monolithic. This corresponds to the *No Selection* baseline in Table 1. We additionally consider variations where the controller backbone to encode images in the support set is either CNN-based or the same slot attention network than the executor.

| Split | No Selection | No Imagination | | With Imagination | |
|---|---|---|---|---|---|
| | Full Slot Att. | CNN/Slot Att. | Full Slot Att. | CNN/Slot Att. | Full Slot Att. |
| Val In-D | $98.9 \pm 0\%$ | $98.3 \pm 0.5\%$ | $98.6 \pm 0.5\%$ | $94.8 \pm 0.5\%$ | $95.4 \pm 3.4\%$ |
| Test OOD | $20.4 \pm 5.7\%$ | $13.1 \pm 5.4\%$ | $34.9 \pm 11.8\%$ | $36.8 \pm 19.8\%$ | $58.6 \pm 25.1\%$ |

Table 1: Accuracies on queries of the validation and OOD test set. Baselines are specified by their controller/executor backbones (Slot Att. = Slot Attention). Results are averaged over five random seeds.

**Results** The results in Table 1 and Figure 2 indicate that (1) object-centric representations are not enough to obtain systematic generalization in a simple visual reasoning task and that the modularity of the mechanisms that operate on slots is important; (2) our proposed compositional imagination framework helps to generate scenarios that were not seen during training and leads to better systematic generalization of the model. On the other hand, we do observe a large variance in terms of OOD performance. One hypothesis is that during imagination *programs* are sampled in a uniform way and are not conditioned on the support set at hand. This unconditioned sampling may lead to invalid programs that the controller cannot infer. How to sample valid program constitutes an interesting future research direction. We provide qualitative results on the selectivity patterns of the proposed method in Appendix A.

## 5 CONCLUSION

We have introduced a new compositional imagination framework to improve the performance of machine learning models in analogical reasoning tasks. We leverage object-centric representations

to decompose problems into a series of objects, properties, and transformations. Once learned, we re-assemble these factors in novel ways, generating new unseen problems on which the model is further trained. We obtain encouraging results on a simplified version of ARC that we will make publicly available to facilitate further research on this challenging benchmark. Future research will focus on multi-step reasoning tasks as well as adapting the model to the original ARC benchmark.

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

## A    SORT-OF-ARC

We introduce Sort-of-ARC, a toy version of ARC that shares the same input space and presents a series of simpler problems. Currently, all the images in Sort-of-ARC are of size $20 \times 20$, each containing 3 objects of size $3 \times 3$. We list them in Figure 3. Each object is defined by a shape, a color and position. There are 16 different shapes and 10 different colors in addition to the background color. The positions are sampled so that none of the objects initially overlap with each other. Each support set is composed of 5 input/output pairs and the model is trained to predict the output of a single query image.

Each episode is constructed by sampling at random a condition and a transformation. The conditions can be either on the shape index or on the color whereas the transformations correspond to simple translations in one of the four cardinal directions. To avoid data points where the output is the same as the input, when generating an episode, we make sure that the condition is satisfied for at least one object of each input image of the support set.

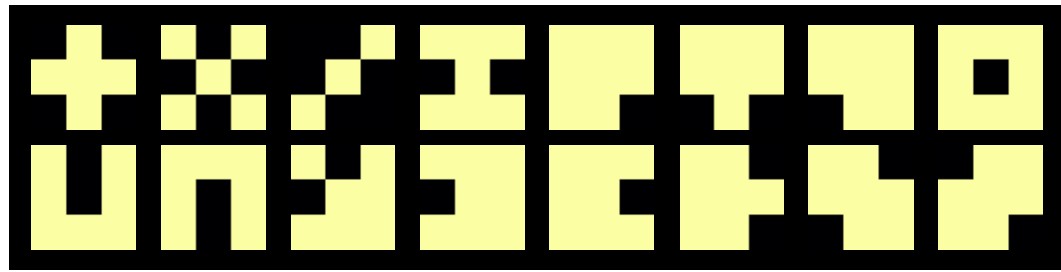

Figure 3: List of object shapes present in the dataset

## B    QUALITATIVE RESULTS

Figure 4 shows the attention maps on the learned conditions and transformations given different input programs. For each ground-truth program (defined by a ground truth condition and transformation) we report, for each partitioned condition (resp. transformation) their proportion of having the maximum probability of being selected.The sparse pattern indicates a clear selectivity for each of the different conditions and transforms (both in distribution and out-of-distribution) depending on the ground-truth program despite some overlaps.

## C    MODEL AND TRAINING DETAILS

The model is trained on 10000 episodes. We distinguish between two phases during training: during the first phase the model is trained on episodes coming from the training set only and on during the second phase, imagined episodes are added and the model is trained to predict the condition/transformation that sampled to produce these episodes. In order for the imagination loss ($\mathcal{L}_{im}$) to be informative enough, we observed that it helped to wait for a few epochs before introducing it, hence the two phases. The imagination loss is introduced after 300 epochs and the corresponding coefficient in increased linearly from 0 to 10 during 200 epochs.

During both phases the model is trained to reconstruct both the query outputs ($\mathcal{L}_{query}$) and the outputs of each input of the support set ($\mathcal{L}_{support}$). We also include some auxiliary loss $\mathcal{L}_{rec}$ to train the perception part of the model to reconstruct images $\mathbf{X}$ from encoded slots $\mathbf{H}$ before any update such that $\mathcal{L}_{rec} = \text{CrossEntropyLoss}(\mathbf{X}, \text{decode}(\text{encode}(\mathbf{X})))$ with $\mathbf{H} = \text{encode}(\mathbf{X})$.

The total loss is thus expressed as :

$$\mathcal{L} = \mathcal{L}_{query} + \mathcal{L}_{support} + \alpha_{rec}\mathcal{L}_{rec} + \alpha_{im}\mathcal{L}_{im}.$$

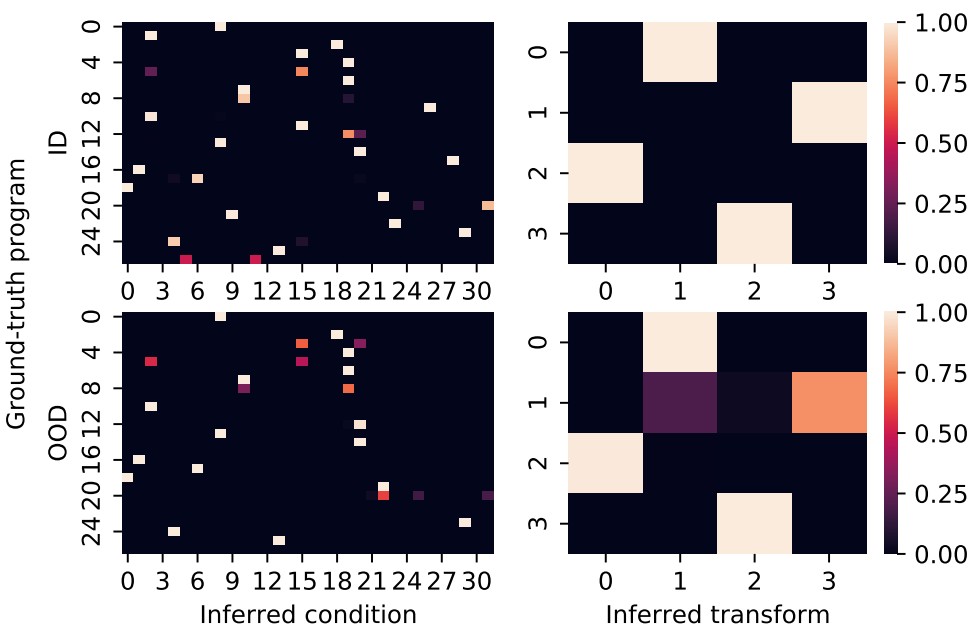

Figure 4: Selection maps. For each ground truth condition (resp. transformation) we report the probability of each condition (resp. transformation) embedding of being selected. Ground truth indices are reported on the y axis and partitioned embedding indices on the x axis.

Following recommendations to train the slot attention model [20], we use a linear learning rate warmup from $1^{-6}$ to $1^{-4}$ over the first 10 epochs. The overall pseudo-code is described in Figure 1 and a sketch of the model is described in Figure 5.

## C.1 MODEL VARIATIONS

**Controller** The controller is composed of a perception backbone than can either be CNN or slot attention-based [20]. The perception backbone is then followed by a transformer encoder [29] and a mean average to aggregate the information from all the elements of the support set.

For the CNN-based backbone, each image (input and output) is independently encoded using a simple CNN. For each pair in a support set we also encode the difference between the output and the input. The input, output and difference embeddings are then concatenated to form the representation of the I/O that will then be processed by the transformer encoder.

For the slot-based backbone, the slot attention module is shared with the perception part of the executor. The input of each I/O pair of a support set is first encoded into a set of slots $\mathbf{H}_{in}$ (starting from random slots) and the output is encoded into a set of output slots $\mathbf{H}_{out}$ using the same slot attention module but starting from the input slots $\mathbf{H}_{in}$. The input and output slots are then concatenated in a slot-wise manner $\mathbf{H}^{\text{pair}} = [\mathbf{H}_{in}, \mathbf{H}_{out}]$ and an I/O pair is represented by a vector $\mathbf{z}_s$ using a weighted sum of each slot contribution such that:

$$\mathbf{z}_s = \sum_{i=1..K} w_i \mathbf{h}_i^s,$$

with the slot-wise importance weight $w_i = f_{\text{importance}}(\mathbf{H}_i^{\text{pair}})$ and the slot-wise contribution $\mathbf{h}_i^s = f_{\text{contribution}}(\mathbf{H}_i^{\text{pair}})$ where both $f_{\text{importance}}$ and $f_{\text{contribution}}$ are parametrized with simple MLPs. Each I/O representation of the support set will also be processed by the transformer encoder to get a final support episode representation $\mathbf{z}$ that will be given to the Executor.

---

**Algorithm 1** Model pseudo-code

---

1: Initialize all parameters
2: Set max_iters and warmup_iters
3: **for** $it \leq$ max_iters **do**
4:     `# Controller Block`
5:     $\{\mathbf{X}, \mathbf{Y}\}, \{\mathbf{x}_q, \mathbf{y}_q\} \sim \mathbf{D}$                   ▷ Sample support set and a query from the dataset
6:     $\mathbf{z} = \text{Controller}(\mathbf{X}, \mathbf{Y})$                   ▷ Obtain program instructions
7:     $\mathbf{H} = \text{Slot\_Attention}(\mathbf{x}_q)$                   ▷ Extract slots
8:     `# Executor block`
9:     $(\mathbf{c}, \mathbf{c}_{prob}), (\mathbf{p}, \mathbf{p}_{prob}) = \text{Selection}(\mathbf{z})$               ▷ Infer neural program
10:     **function** UPDATE($\mathbf{H}, \mathbf{c}, \mathbf{p}$)
11:         $\alpha_k = MLP_{pres}([\mathbf{h}_k, \mathbf{c}])$        ▷ Predict the presence of each slot for input update
12:         $\tilde{\mathbf{h}}_k = MLP_{update}([\mathbf{h_k}, \mathbf{p}])$            ▷ Obtain update for each slot
13:         $\mathbf{h}_k^{new} = \mathbf{h}_k + \alpha_k * \tilde{\mathbf{h}}_k$           ▷ Update slots according to presence
14:         **return** $\mathbf{H}^{new}$
15:     **end function**
16:     $\mathbf{H}_k^{new} = \text{Update}(\mathbf{H}, \mathbf{z})$
17:     $\mathbf{o} = \text{Decode}(\mathbf{H}^{new})$                   ▷ Decode updated slots
18:     $\mathcal{L}^{rec} = \text{cross\_entropy}(\mathbf{o}, \mathbf{y}_q)$           ▷ Compute reconstruction loss
19:     `# Imagination`
20:     **if** $it \geq$ warmup_iters **then**
21:         $i_c, i_p \sim U(1, N_c), U(1, N_p)$           ▷ Randomly sample a neural program
22:         $\mathbf{o}_i^{i_c, i_p} = \text{Update}(\mathbf{H}_i, \mathbf{c}_{i_c}, \mathbf{p}_{i_p})$ ▷ Imagine new support outputs by applying the randomly sampled program to the original inputs
23:         $\{\mathbf{X}, \hat{\mathbf{Y}}\} = \{\mathbf{x}_i, \mathbf{o}_i^{i_c, i_p}\}_{i=1..N}$         ▷ Build a new support set with imagined outputs
24:         $\mathbf{z}^{i_c, i_p} = \text{Controller}(\mathbf{X}, \hat{\mathbf{Y}})$           ▷ Obtain program instructions
25:         $\mathbf{c}_{prob}^{i_c, i_p}, \mathbf{p}_{prob}^{i_c, i_p} = \text{Selection}(\mathbf{z}^{i_c, i_p})$        ▷ Infer randomly sampled neural program
26:         $\mathcal{L}^{im} = -\log(\mathbf{c}_{prob}^{i_c, i_p})_{i_c} - \log(\mathbf{p}_{prob}^{i_c, i_p})_{i_p}$       ▷ Compute imagination loss
27:         $\mathcal{L}^{total} = \mathcal{L}^{rec} + \mathcal{L}^{im}$         ▷ Add imagination loss to reconstruction loss
28:     **else**
29:         $\mathcal{L}^{total} = \mathcal{L}_{rec}$
30:     **end if**
31: **end for**

---

**No selection** For the baseline without any selection bottleneck the slots are still updated in an entity-centric manner but this time the output of the controller $\mathbf{z}$ is directly given to the updating module such that, following the same notations introduced previously, a slot $\mathbf{h}_i$ is updated as :

$$\mathbf{h}_i^{\text{new}} = \mathbf{h}_i + g(\mathbf{h}_i, \mathbf{z}),$$

with g parametrized by a simple MLP.

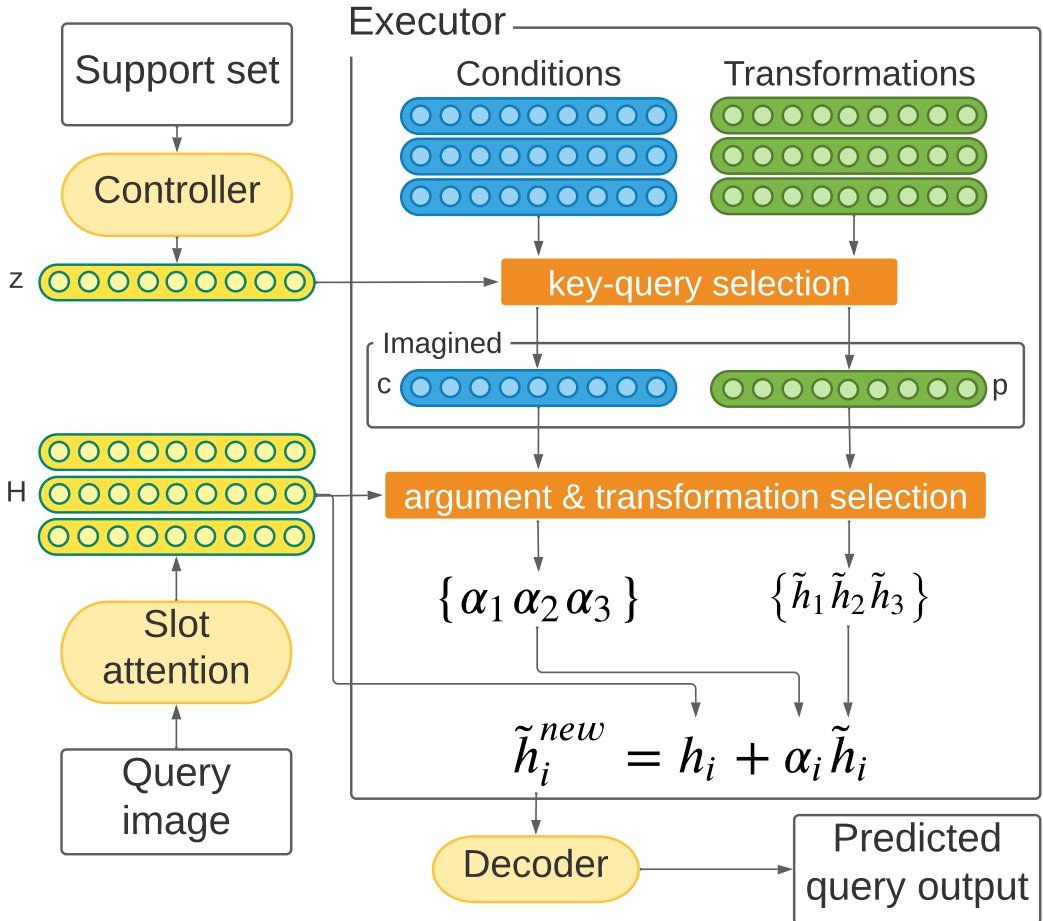

Figure 5: Model architecture

