# OpenReview forum: "Object-centric Compositional Imagination for Visual Abstract Reasoning"
_ICLR.cc/2022/Workshop/OSC — ICLR2022 OSC  Poster_

### Official Review · Reviewer_AwqN · 2022-03-15
**Nice workshop paper with insightful contribution but limited experiments**

**Rating:** 2
**Confidence:** 3

**Review:**

### Summary
This paper proposes an object-centric approach to "imagination" where a compositional inductive bias in the architecture is leveraged to generate new data points that are novel combinations of known concepts. Training models of such imagined scenarios helps them generalize more systematically.

The setting explored here is a simplified version of the Abstraction and Reasoning Corpus (ARC), named Sort-of-ARC, that uses the same input space, but considers a simpler (more restricted) set of operations to map inputs to outputs. In particular, the model is given access to a support set of correct input/output pairs, where the outputs are the result of applying a transformation (eg. spatial translation) to the set of objects contained in the inputs that satisfy some condition (eg. all red objects). To succeed at this task in manner that leads to systematic generalization, the model first has to infer the underlying program that generated the support set and then apply it to the query input to generate the right output. This is analogous to other abstract visual reasoning settings, such as Raven's progressive matrices.

The proposed model consists of a controller and an executor. The controller encodes the input/output images in the support set to produce a set of latents, called the instruction embedding, to give to the executor. The executor decompose the query input to obtain a set of entity-centric latent representations and updates them based on a neural program, after which they are decoded. The neural program is obtained by using the instruction embedding to query from a set of learned condition and transformation embeddings. The program is applied separately to each entity embedding to update them. Here the condition embedding gates the proposed update produced by the transformation to select only relevant entities to which the program applies.

Compositional imagination can now acts as a learning paradigm, through applying arbitrary condition-transformation pairs to the inferred entities of the inputs from the support set to produce new outputs (this amounts to sampling indices for these embeddings, and applying the executor), The controller can then be trained on this new support set to yield program instructions that select condition and transformation embeddings that were used. This imagination loss is combined with the regular query output prediction loss.

The experiments indicate that (1) it is important that the mechanisms that operate on the object representations are modular themselves, and (2) that training with the imagination loss improves out of distribution generalization.

### Review

This is a nice workshop paper that explores compositional imagination as a framework for out of distribution generalization. The setting considered requires novel compositions to obtained through applying (condition, transformation) pairs, where it is further demonstrated that modularity at the level of mechanisms is also important. The proposed architecture for solving these tasks is intuitive, although perhaps more development could go into the design of the controller to produce the instructions. The novelty of the proposed mechanisms is limited, but this is okay. The paper is quite clear and easy to read. Some pro's and con's:

pro's

* interesting results on novel benchmark, promising step towards ARC
* compositional imagination is an interesting training paradigm adapted from prior work to the object-centric setting

con's

* experiments indicate benefit of modular mechanism application, but the OOD results are not fully convincing yet. A potential reason for this is given by not explored
* additional exploration regarding why selection hurts within distribution generalization or further exploration of imagination-based training (when it works, when it doesn't) would be helpful.

With regards to further developing this work I would encourage the authors to consider different modes of OOD generalization, eg. as considered in [2].

---

### Official Review · Reviewer_BAgy · 2022-03-17

**Rating:** 2
**Confidence:** 2

**Review:**

This paper proposes the integration an imagination loss onto the typical entity centric setting and fits well with the theme of the workshop. At the same time though -- I think the paper could be strengthened in the following ways:

Can the authors show the systematic generalization of their approach? What do the in distribution and out of distribution datasets correspond to? Can the authors qualitatively show the imaginations from their approach? Can they qualitatively show the difference with prior approaches?

What is the underlying principle for doing the imagination setting the authors propose? The overall approach seems rather heuristic to me.

---

### Official Review · Reviewer_Txgz · 2022-03-19
**Promising methodology of a challenging application**

**Rating:** 2
**Confidence:** 3

**Review:**

# Summary

This paper proposes task imagination mechanism at the top of object-centric representation learning while also considering the scenario of out-of-distribution adaptation. The whole scheme consists of several steps:

- A conditional model (controller) that generates a task embedding based on the given datasets.
- A object-centric encoder where the k-slot representations will be updated according to the task embedding given by the controller.
- An imagination mechanism to generate unseen tasks based on the combination of learned `condition c` and `transform p`.

In my opinion, the methodology is very promising. Also the application (ARC) is quite special since the image pairs represent some underlying transformation, and it is more difficult than image reconstruction task in some recent object-centric learning works. I vote for acceptance.

# Some Questions
Probably I miss something. There are also some unclear parts in the experiments that may need some explanation.
1. How the hyperparams are set (e.g. $\alpha_{rec}$, $\alpha_{im}$ in the loss, the total number $c$ and $p$.)
2. What's the difference between query output loss and support set output loss.
3. Do we really need both `condition c` and `transform p` to modulate the object-centric representations? A ablation comparison against using only one (i.e. $h_k^{new} = h_k + \tilde{h}_k$) will be convincing.
4. More explanation is need for the metrics in Table 1:
    - Although accuracy is improved for OOD tasks, the performance of *With Imagination* has $~3$% drop for in-domain tasks. Is it caused by the suboptimal setting of the regularization coefficient ($\alpha_{im}$)? If so, a hyperparameter search could be helpful to provide more insights.
    - The performance of *With Imagination* and *No Imagination* in OOD tasks is not convincing for me. Considering the super large variance of  *With Imagination*, it is hard to say it is better. Nevertheless, the advantage of the other components is obvious.

---

### Decision · Program_Chairs · 2022-03-24

Accept (Poster)